# Offline ventral subiculum-ventral striatum serial communication is required for spatial memory consolidation

G. Torromino[1], L. Autore[1], V. Khalil[1], V. Mastrorilli[1], M. Griguoli [2], A. Pignataro[3,4], E. Centofante[1], G.M. Biasini[1], V. De Turris [5], M. Ammassari-Teule[4], A. Rinaldi[1] & A. Mele[1]*

The hippocampal formation is considered essential for spatial navigation. In particular, subicular projections have been suggested to carry spatial information from the hippocampus to the ventral striatum. However, possible cross-structural communication between these two brain regions in memory formation has thus far been unknown. By selectively silencing the subiculum–ventral striatum pathway we found that its activity after learning is crucial for spatial memory consolidation and learning-induced plasticity. These results provide new insight into the neural circuits underlying memory consolidation and establish a critical role for off-line cross-regional communication between hippocampus and ventral striatum to promote the storage of complex information.

[1] Department of Biology and Biotechnology 'C. Darwin', Center for Research in Neurobiology 'D. Bovet', Sapienza University of Rome, Rome, Italy. [2] European Brain Research Institute (EBRI), Rome, Italy. [3] CNR - National Research Council, Institute of Translational Pharmacology, Rome, Italy. [4] Laboratory of Psychobiology, Department of Experimental Neurosciences, IRCCS Santa Lucia Foundation, Rome, Italy. [5] Center for Life Nanotechnology - Nanotechnology for Neuroscience, Istituto Italiano di Tecnologia (IIT), Rome, Italy. *email: andrea.mele@uniroma1.it

Established models describing stabilization of episodic memory suggest that the hippocampus (HPC) plays a crucial role in the early stages of memory consolidation (see ref. [1]). This view has recently evolved with the demonstration that memory consolidation depends upon the coordinated activity of the HPC and a network of brain regions including the neocortex and key subcortical structures[2–4]. In this framework, neural activity in the ventral striatum (VS) has recently emerged as a fundamental event supporting spatial learning and memory consolidation. Indeed VS post-training manipulations induce deficits in spatial memory tasks, suggesting the necessity of off-line neural activity within this brain region in order to consolidate spatial memories[5–8]. In line with these findings, it has been demonstrated that key molecular alterations, required for successful consolidation of newly acquired spatial information in the HPC, also occurs in the VS[9].

The VS receives prominent glutamatergic projections from the prefrontal cortex, the amygdala and the HPC formation[10–13]. It has been hypothesized that each one of these distinct pathways relays specific information to the VS. In particular, electrophysiological recordings in rodents suggest that HPC–VS projections might carry spatial information[14,15]. Disconnection experiments provide further support for a cross-regional interplay between HPC and VS in spatial and contextual information processing[16–18]. The fact that these manipulations were performed pre-training, however, leaves uncertainty on as whether disruption of HPC–VS connections impairs encoding of spatial information or disrupts the consolidation of regularly encoded spatial information.

Relevant electrophysiological evidence showing off-line coherent reactivation of neuronal ensembles in the HPC and in the VS after learning[19], supports the hypothesis that post-training cross-structural communication between these two brain structures might be a critical mechanism for long-term stabilization of spatial information. However, the causal role of the functional interplay between HPC and VS in memory consolidation remains to be proven. In this study by silencing subicular projections to the VS, we demonstrated that off-line serial communication between these two brain regions is needed for spatial memory consolidation and the development of learning induced structural changes necessary for long-term memory stabilization.

## Results

**Learning-induced activation of ventral subiculum–VS projections**. To provide direct evidence supporting the hypothesis that HPC–VS communication is required for spatial memory consolidation, we started exploring spatial learning-induced neuronal activation in HPC projections to VS, combining retrograde tracing with cell-activity dependent labeling in mice trained in the spatial version of the Morris water maze (sMWM). One week after Fluoro-Gold (FG) administrations in VS, mice were trained in the sMWM using a single-day massed procedure. Control groups were either naïve to training or trained in the cue version of the maze (cMWM) (Fig. 1a). Retrograde FG tracing confirmed the ipsilateral ventral subiculum (vSUB) as the major source of HPC projections to the VS (Fig. 1b; Supplementary Fig. 1b, c)[10]. A few labeled neurons were found also in the dorsal subiculum and the ventral CA1 (Fig. 1b). Importantly, we found no contralateral subicular projections (Supplementary Fig. 1b). Interestingly, fos expression was significantly increased in the entire vSUB, as well as in the subicular VS-projecting cells, after both cue and spatial learning compared to naïve mice (Fig. 1c, d) indicating that vSUB is activated regardless of the navigational strategy[20].

**Off-line vSUB-VS disconnection impairs spatial memory**. To verify the causal relationship between the neuronal activation

observed in the vSUB–VS pathway and spatial memory consolidation, we carried out a pharmacological asymmetrical disconnection between the two structures immediately after learning (Fig. 2a). The rationale underlying our use of an asymmetrical disconnection is based on the assumption of a serial transfer of information between the two brain regions and lack of contralateral connections between them[18,21]. If the information needed to solve the task is transferred from the vSUB to the VS, unilateral lesion of the vSUB prevents the intact VS in the same hemisphere to access this information while, in the contralateral hemisphere, unilateral lesion of the VS prevents this region to process the information sent by the intact vSUB. Thus, potential memory impairing effects of this asymmetrical disconnection have to be ascribed to disruption of serial information processing. As a preliminary experiment, we verified the effects of bilateral inactivation of the vSUB or the VS in the sMWM. In both cases, immediate post-training administration of the NMDA antagonist, AP-5, impaired mice performance on the probe trial 24 h after training (Supplementary Fig. 2). To test the role of the cross-regional communication, an independent group of animals was administered AP-5 unilaterally in the vSUB and contralaterally in the VS, immediately after the end of training in the sMWM (Fig. 2a). The post-training functional disconnection between the two brain structures impaired the ability to correctly locate the platform 24 h after training, compared to vehicle controls (Fig. 2a; Supplementary Fig. 3a, c). On the contrary, a control group receiving the NMDA antagonist ipsilaterally in the two brain regions showed an intact ability to reach the platform compared to vehicle-injected mice (Fig. 2b; Supplementary Fig. 3d, f). To test the generality of this finding, we assessed the effects of the off-line vSUB–VS functional disconnection on spatial memory performance in an object displacement task (ODT) (Fig. 2c). As previously observed in the sMWM, the AP-5 disconnected group showed an impaired ability to discriminate the spatial change compared to vehicle-injected controls (Fig. 2c; Supplementary Fig. 5).

**vSUB-VS activation is not necessary for cue-memory**. Cell activity-dependent labeling in the vSUB–VS projection after cue learning suggests a role of the pathway independent of the type of information being consolidated. To verify this possibility, an additional control group was injected in the vSUB and in the contralateral VS immediately after training in the cMWM. Mice administered AP-5 were able to locate the platform 24 h after training in a manner similar to vehicle controls (Fig. 2d; Supplementary Fig. 6), also suggesting that the effect observed in the sMWM cannot be attributed to differences in motivational processes. Overall these findings demonstrate the need for intact neuronal activity in both vSUB and VS to ensure spatial memory consolidation. Most importantly, they demonstrate for the first time that post-training serial communication between these two brain regions is specifically needed to stabilize spatial information early after encoding.

**vSUB is required for spatial learning-induced VS plasticity**. We have recently demonstrated that increased spine density in the VS accompanies the consolidation of spatial memories[22]. Having shown that off-line interference with communication between the vSUB and the VS impairs spatial memory, we wondered whether there was a role for this connection in learning-induced structural plasticity in the VS. Mice were trained in the sMWM and immediately post-training injected unilaterally with AP-5 in vSUB. The contralateral vSUB was injected with vehicle as an internal control (Fig. 3a; Supplementary Fig. 8). To verify the specificity of the effect, further groups of animals were trained in the cMWM (Supplementary Fig. 8). A group of naïve mice was used to normalize data obtained from cue and spatial trained

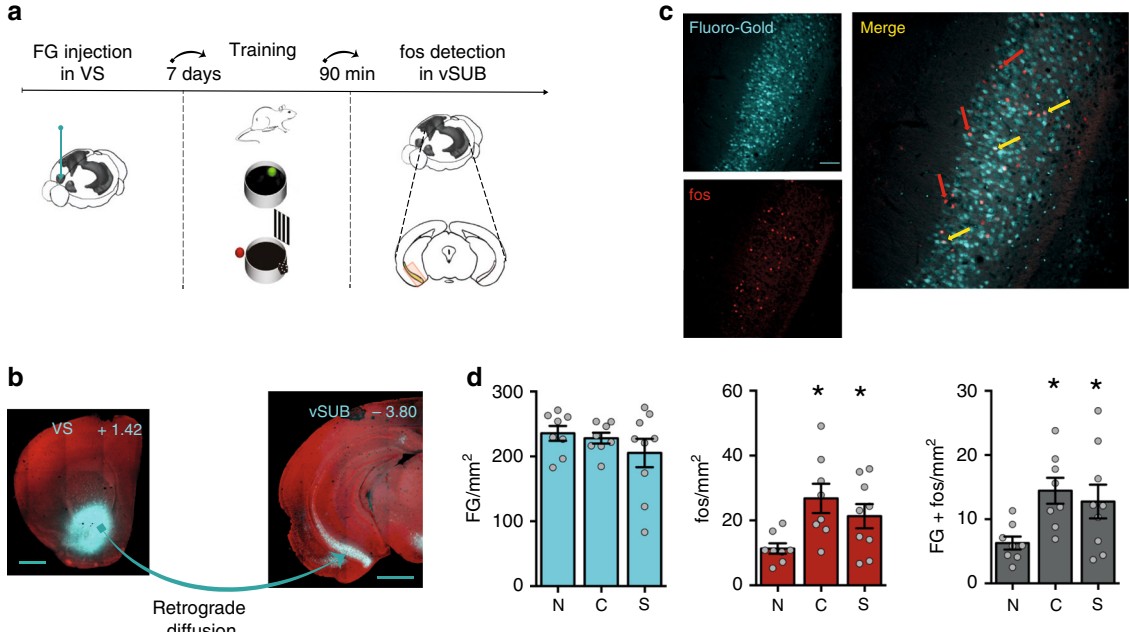

**Fig. 1 vSUB projection to the VS is activated by training in the water maze. a** Schematic of the experimental design. Retrograde tracer Fluoro-Gold (FG) combined with fos was used to identify subicular VS-projecting cells activated by cue or spatial training in the MWM. MWM images modified from ref. [9]; Image credit for brains schematics: Allen Institute. **b** Microphotographs showing FG injection site in the VS and VS projecting cells in the vSUB (cyan). Images are counterstained with the nuclear dye propidium iodide (red). Scale bar: 1 mm; coordinates from bregma (mm). **c** Representative images showing FG, fos labeling and the overlay of the two labeling in the vSUB (red arrows: fos; yellow arrows: co-labeling). Scale bar: 100 μm. **d** fos expression was significantly increased in the whole vSUB ($F_{2,22} = 4.762$, $p = 0.0191$), as well as in the subicular VS-projecting cells ($F_{2,22} = 4.178$, $p = 0.0290$), in both cue (C; $n = 8$) and spatial (S; $n = 9$) trained mice compared to naïve (N = naïve to training; $n = 8$). No difference was found among groups in the number of FG-positive cells ($F_{2,22} = 1.023$, $p = 0.3762$). Histograms represent mean ± SEM. *$p < 0.05$ vs naïve (Fisher).

mice. Spine counts in VS 24 h after training demonstrated a spatial learning-induced specific increase in spine density, that was impaired by blockade of vSUB activity (Fig. 3b). This result confirms our early findings of increased spine density in VS after spatial learning[22], and validates the hypothesis that vSUB neuronal activity is required to promote learning-induced structural changes in the VS.

**Inhibition of vSUB–VS projections impairs spatial memory.** To determine whether the effect of vSUB–VS pathway disconnection observed was due to direct projections from the vSUB to the VS, we used the hM4D$_i$–clozapine-N-oxide (CNO) chemogenetic neural inhibition tool to rapidly and selectively inhibit vSUB–VS projecting neurons (Fig. 4a, e). We first examined whether activation of hM4D$_i$ in VS-projecting subicular terminals by local CNO delivery could modulate the activity of VS neurons by recording spontaneous excitatory postsynaptic currents (sEPSCs) from medium spiny neurons (MSNs) in VS slices of a cohort of mice ($n = 6$) injected with the viral vector in the vSUB (Fig. 4a). In 10 out of 13 cells recorded, bath-application with 10 μM CNO decreased the frequency (Fig. 4b, c) but not the amplitude of sEPSCs (Fig. 4b, d), indicating that hM4D$_i$ DREADDs activation in VS-projecting subicular terminals decreased glutamate release onto MSNs in the VS. The latency of the CNO-mediated effect on the EPSCs frequency was $9.30 \pm 1.1$ min. The analysis of the resting membrane potential (Vm), the input resistance ($R_{input}$) and the firing frequency (in response to depolarizing current steps; 800 ms duration, 300 pA amplitude) did not show differences before and during CNO application (Supplementary Fig. 9a–d). CNO application did not affect either the frequency or the amplitude of EPSCs (Supplementary Fig. 9e, g) recorded in spiny neurons ($n = 8$) obtained from naïve mice ($n = 5$).

Next a cohort of mice was bilaterally implanted with guide cannulae above the VS after viral vector injection in the vSUB (Fig. 4e), to allow the selective manipulation of the VS-targeting subicular fibers (Fig. 4f; Supplementary Fig. 10). All mice underwent training in the sMWM (Supplementary Fig. 11a, b) and were administered with vehicle or CNO 10 mM[23] immediately after training. The probe test performed 24 h later revealed that CNO-injected mice spent significantly less time in the target quadrant compared to vehicle-injected mice (Fig. 4g). To establish the temporal dynamic of the involvement of vSUB–VS projection in spatial memory consolidation, we performed an additional experiment administering CNO in the VS 120 min after training. Both vehicle and CNO administered mice were able to correctly locate the platform (Fig. 4h; Supplementary Fig. 11d–f), thus demonstrating that the activity of this pathway has a time-limited role for spatial memory consolidation. VS CNO administrations in independent groups of mice demonstrated that activation of hM4D$_i$ in VS-projecting subicular terminals was void of significant effects on anxiety, locomotion and novelty (Supplementary Fig. 12). Further control experiments ruled out also possible effects of CNO alone on learning, locomotor activity or anxiety (Supplementary Fig. 13). These data confirmed the need for post-training cross-regional communication between the vSUB and the VS to consolidate spatial information, and demonstrated also that this effect is mediated by direct projections between the two structures. Overall, our results reveal that spatial memory consolidation and related structural plasticity depend upon off-line communication between the vSUB and the VS in the early stages after learning.

**Discussion**
Focusing on the early stages of memory consolidation, our findings establish that neural communication between HPC and

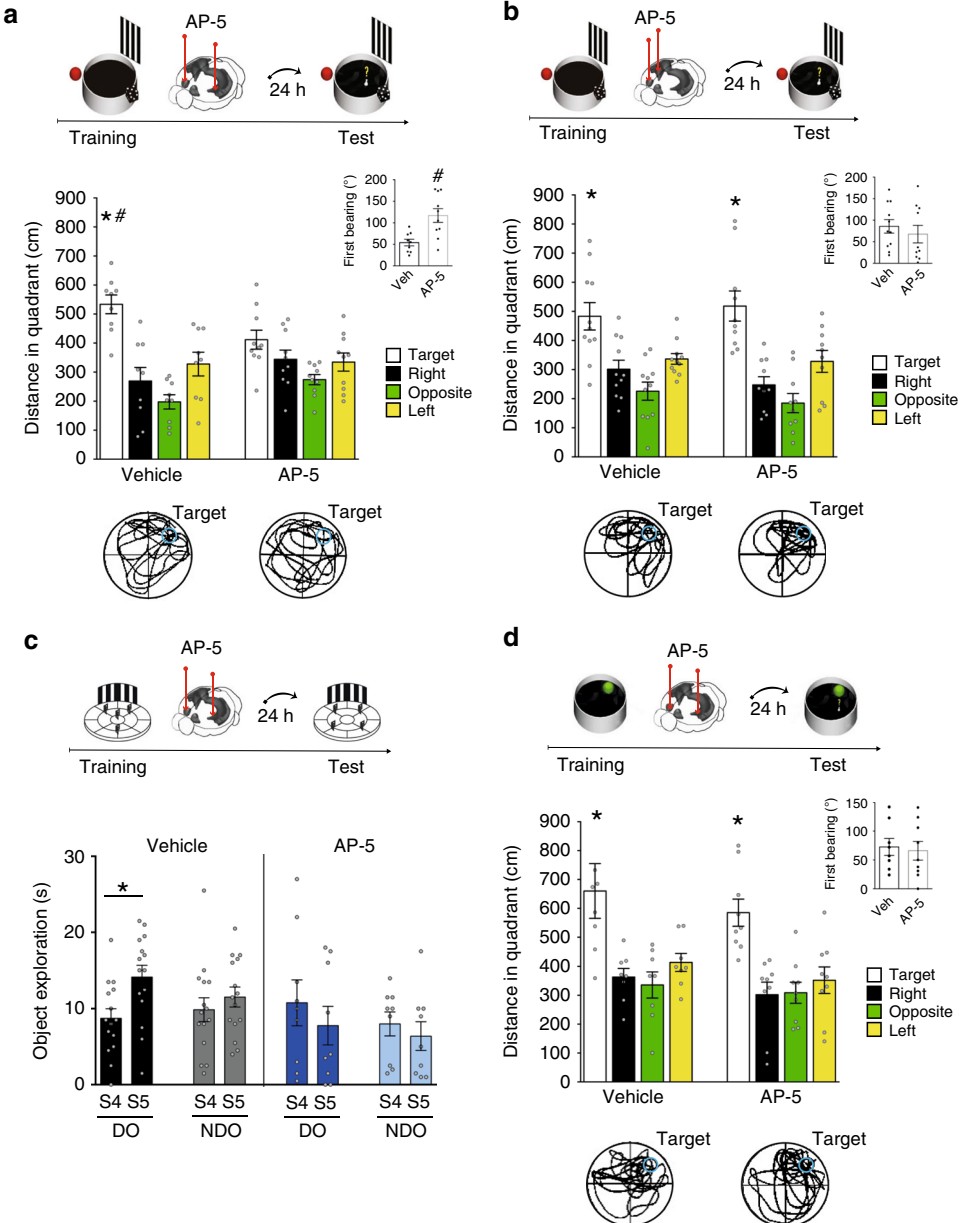

**Fig. 2 vSUB–VS pathway disconnection impairs spatial memory consolidation. a** Immediate post-training functional disconnection between the vSUB and the VS with AP-5 ($n = 10$), impaired spatial memory consolidation compared to vehicle controls ($n = 9$) (two-way ANOVA of quadrant preference $F_{3,51} = 15.824$; $p = 0.0001$; treatment $F_{1,17} = 0.270$; $p = 0.6100$; quadrant × treatment $F_{3,51} = 4.104$; $p = 0.0110$). The insert shows first bearing of mice in the two groups ($t_{17} = 3.450$; $p = 0.0031$; unpaired t-test). Bottom panels: representative path from vehicle and AP-5 animals. **b** Unilateral post-training AP-5 ($n = 10$) administrations in the vSUB and in the ipsilateral VS did not impair mice ability to locate the platform compared to vehicle-injected mice ($n = 11$) (two-way ANOVA of quadrant preference $F_{3,57} = 20.775$, $p = 0.0001$; treatment $F_{1,19} = 1.077$, $p = 0.3124$; quadrant preference × treatment $F_{3,57} = 0.499$, $p = 0.6846$). The insert shows first bearing of mice in the two groups ($t_{19} = 0.709$; $p = 0.4871$; unpaired t-test). Bottom panels: representative path from vehicle and AP-5 animals. **c** Immediate post-training vSUB–VS functional disconnection ($n = 9$) impaired mice ability to discriminate spatial displacement 24 h after training, compared to vehicle-injected mice ($n = 15$) (vehicle: session × objects $F_{1,8} = 5.34$, $p = 0.036$; AP-5: session × objects $F_{1,14} = 0.37$; $p = 0.55$). Histograms represent mean ± SEM; *$p < 0.05$ displaced objects (DO) in S5 vs DO in S4 (within group, Fisher). **d** Post-training functional disconnection between vSUB and VS ($n = 9$) did not affect mice ability to locate the platform 24 h after training in the cMWM (vehicle $n = 8$) (two-way ANOVA of quadrant preference $F_{3,45} = 14.814$; $p = 0.0001$; treatment $F_{1,15} = 1.134$; $p = 0.3037$; quadrant preference × treatment $F_{3,45} = 0.369$; $p = 0.7754$). The insert shows first bearing of mice in the two groups ($t_{15} = 0.299$; $p = 0.7692$; unpaired t-test). Bottom panels: representative path from vehicle and AP-5 animals. Histograms represent mean ± SEM; *$p < 0.05$ target vs right, opposite, left (within group); #$p < 0.05$ target vs target (between groups, Tukey). MWM images modified from ref. [9]; Image credit for brains schematics: Allen Institute.

VS is required immediately after learning to properly store spatial information and to promote in the VS the structural changes underlying it.

In particular, the experiments presented provide direct indication that activity in the two brain regions is not occurring independently or simultaneously, but rather neural activity in the vSUB is driving neuronal activation in the VS and this cross-structural interaction is a key step for spatial memory consolidation. It is worth noting that in all the experiments presented, we performed post-training manipulations, thus

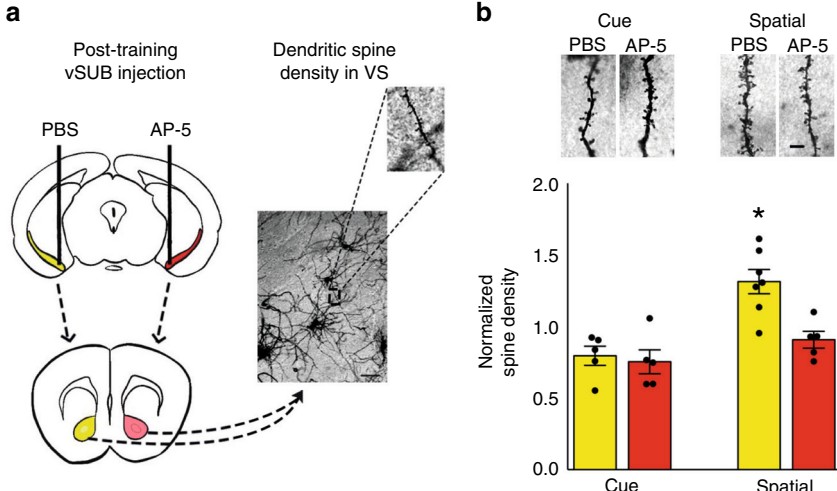

**Fig. 3 Spatial learning-induced increase in VS spine density is vSUB-dependent. a** Schematic representation of the experimental design. Scale bar: 50 μm. **b** Representative microphotographs of dendritic segments of a MSN in the VS for each experimental condition. Mean dendritic spine density/ micrometer for cue and spatial trained mice was normalized to naïve mice ($n = 6$). Spatial training significantly increased spine density in the VS ipsilateral to the vehicle-injected vSUB ($n = 7$) as compared to cMWM ($n = 5$) trained mice. No change was observed in the AP-5 administered hemisphere ($n = 5$) (two-way ANOVA of training $F_{1,18} = 18.162$, $p = 0.0005$; treatment $F_{1,18} = 8.056$, $p = 0.0109$; training × treatment $F_{1,18} = 5.329$, $p = 0.0331$). Scale bar: 5 μm. Histograms represent mean ± SEM; *$p < 0.05$ spatial vs cue (Tukey).

suggesting that active communication between the two brain regions is needed after training. Our data are consistent with prior correlative evidence showing that memory in the early stages of consolidation is sustained by replay in connected brain areas[19] and provide support to a causal role of this communication to long-term storage of spatial information.

The VS receives glutamatergic projections from the HPC, the amygdala and the prefrontal cortex[13,24–26]. This circuit involves also trans cortical networks connecting the HPC to the VS indirectly through the amygdala or the prefrontal cortex[27–29]. Interestingly, activation of different projections leads to distinct physiological and behavioral responses[16,18,30,31]. Different aspects of spatial navigation seem to depend on the activity of separate limbic-cortical-striatal circuits[18]. Direct projection from the HPC to the VS would guide navigation in novel environment independent of previous knowledge. On the contrary, indirect projection through the prefrontal cortex would guide prospective navigation dependent on previously acquired information[18]. Our chemogenetic data challenge this view by demonstrating that consolidation of spatial memory, necessary to perform a delayed recall, depends on direct projection from the vSUB to the VS.

Current models support the occurrence of changes in neuronal morphology and connectivity after spatial learning in the HPC and at later interval in the neocortex[32,33]. Our findings demonstrate that such changes, early after encoding, occur also in the VS, are specifically induced by spatial training and that immediate post-training activity of the subicular input is a necessary requirement. These observations, further support a specific role of the VS and the subicular projection to this structure in explicit memory. They also demonstrate the occurrence of such changes in brain regions outside the medial temporal lobe immediately after the learning experience. Although to our knowledge, demonstration of spatial training-induced structural plasticity outside the medial temporal lobe has been shown only in the VS[22], it would not be surprising if it was occurring in a wider array of brain structures. This would be coherent with mapping studies suggesting that long-term memory relays on the collective activity of a distributed network of cortical and subcortical regions[2,3,34]. Relevant to this issue is the timing of such changes, further studies will be needed to assess their stability over time.

By showing that immediately after learning, off-line serial neural communication between the HPC and the VS is required for the storage of a spatial, but not a cue-based memory, our data identify in the offline HPC-VS interplay a dynamic serial interaction between the two regions specific to spatial memory consolidation. The VS is generally viewed as the downstream structure performing action selection based on HPC and prefrontal cortical input[35–38]. Recent network analysis, on the contrary, locate the VS upstream the HPC and the prefrontal cortex, playing a central role in updating memories based on new information[3]. Although the present study did not specifically investigate this issue, demonstrating control of vSUB projections over VS in spatial memory consolidation, it seems to support the first hypothesis. Moreover, it may represent a mechanism by which VS gates inappropriate responses based on previous experience[38]. Nevertheless, regardless the models considered, our results demonstrate the necessity for VS activity in HPC-dependent memories and a possible mechanism by which different parts of the brain integrate their activity to store relevant information.

## Methods

**Subjects**. All experiments were conducted in naïve CD1 male mice (Charles River, Italy), 10–15 weeks old and weighing 35–50 g at the start of the experiments. Animals were housed in groups of four in standard cages (26.8 × 21.5 × 14.1 cm) with enrichment conditions, water and food ad libitum, under 12 h light/dark cycle and constant temperature (22 ± 1 °C). Behavioral training and testing were conducted during the light period (from 9:00 am to 5:00 pm). All the animals were treated in respect to current Italian and European laws for animal care, and the maximum effort was made to minimize animal suffering.

**Stereotaxic surgery**. Each mouse was deeply anesthetized with chloral hydrate (500 mg/kg; Sigma Aldrich, Italy), and secured on a stereotaxic apparatus (David Kopf Instruments, USA). Two stainless-steel guide cannulae (0.50/0.25 × 7 mm; Unimed, Switzerland) were implanted as follows, depending on the experiment: (1) bilaterally in vSUB; (2) bilaterally in the VS; (3) contralaterally in one vSUB and one VS; (4) ipsilaterally in one vSUB and one VS. Cannulae implantation was performed through craniotomies on the skull at the following coordinates relative to bregma: antero-posterior (AP) = −3.4 mm, medio-lateral (ML) = ± 3 mm, dorso-ventral (DV) = −1.3 mm from bregma for the vSUB; AP = + 1.7 mm, ML = ± 1 mm, DV = −1.3 mm from bregma for the VS. Guide cannulae were anchored to the skull through the use of dental cement (Ilic, Italy). Mice were allowed to recover for at least 1 week after surgery.

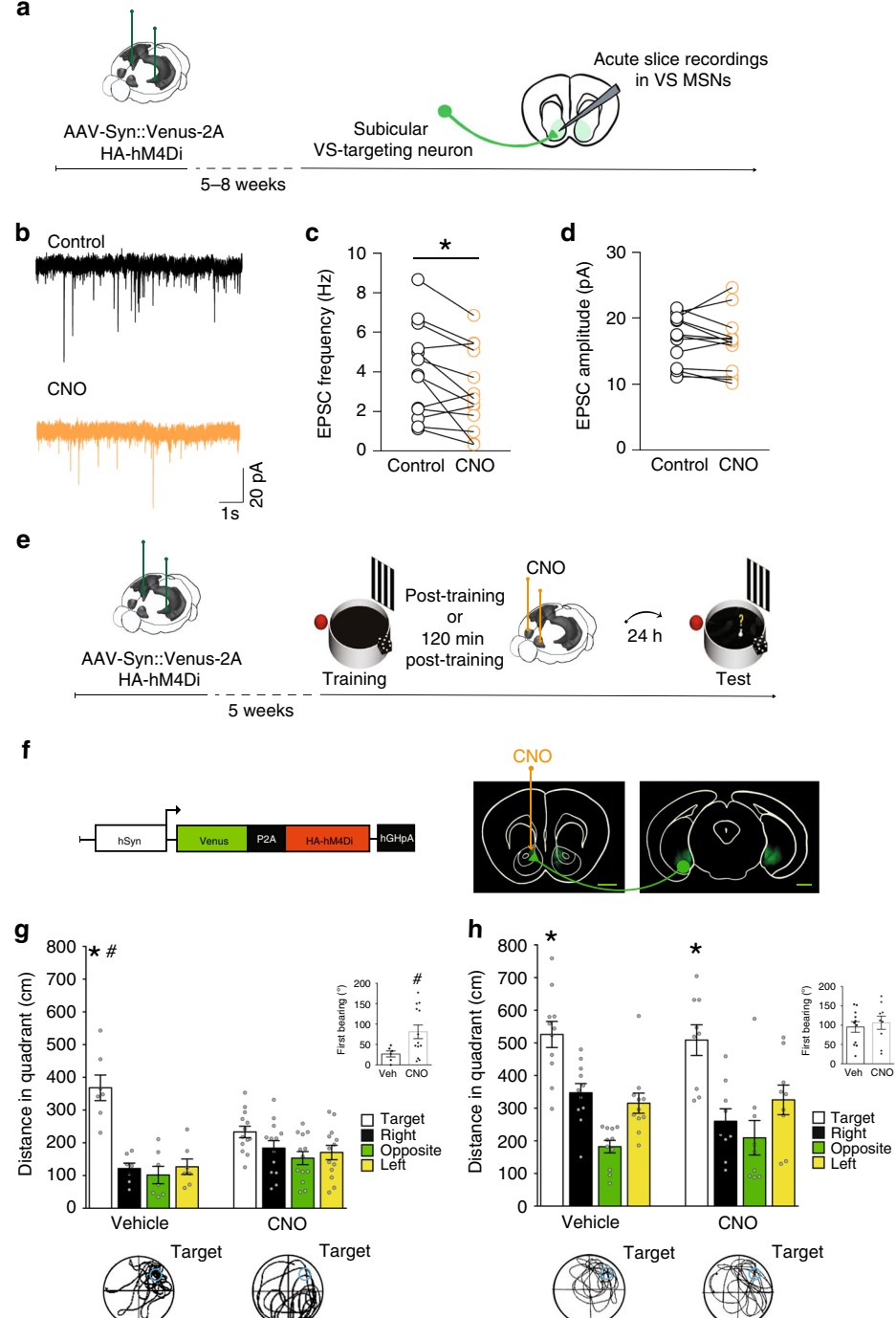

**Fig. 4 Inhibition of vSUB terminals targeting the VS impairs spatial memory consolidation. a** Schematic of the ex vivo electrophysiology DREADDs experiment. **b** Representative traces from VS MSNs showing sEPSCs in control conditions (black) and under CNO (orange) **c** Scatter plot graph summarizing sEPSC frequency in control conditions (black) and under CNO (orange) ($n = 13$ cells/6 mice; $t_{12} = 2.8$, $p = 0.015$; paired $t$-test). **d** CNO did not change the sEPSCs amplitude ($p = 0.38$; paired $t$-test). **e** Schematic of the behavioral DREADDs experiment. **f** Schematic of AAV expressing Venus and the HA-tagged hM4D$_i$ under the h-Synapsyn (Syn) promoter. Representative coronal sections from HPC and VS. Scale bars: 1 mm. **g** Mice infected in the vSUB with the AAV and injected post-training with CNO (10 mM) in the VS showed impaired performance on the probe test. Histograms represent the mean of distance traveled in each quadrant for vehicle ($n = 7$) and CNO ($n = 13$) injected groups (repeated measures ANOVA of quadrant preference $F_{3,54} = 22.976$, $p = 0.0001$; treatment $F_{1,18} = 0.085$, $p = 0.7735$; quadrant preference × treatment $F_{3,54} = 8.019$, $p = 0.0002$). The insert shows first bearing of mice in the two groups ($t_{17} = 3.450$; $p = 0.0031$; unpaired $t$ test). Bottom panels: representative path from vehicle and CNO animals. **h** Mice bilaterally infected in the vSUB and injected 120 min post-training with CNO (10 mM) in the VS showed no impairment in the performance. Histograms represent the mean of distance traveled in each quadrant for vehicle ($n = 11$) and CNO ($n = 9$) injected groups (repeated measures ANOVA of quadrant preference $F_{3,54} = 24.6954$, $p < 0.0001$; treatment $F_{1,18} = 0.4137$, $p = 0.3721$; quadrant preference × treatment $F_{3,54} = 0.8742$, $p = 0.46022$). The insert shows first bearing of mice in the two groups ($t_{18} = 0.5077$; $p = 0.6178$; unpaired $t$-test). Bottom panels: representative path from vehicle and CNO animals. Histograms represent mean ± SEM. #$p < 0.05$ target vs target (between groups); *$p < 0.05$ target quadrant vs right, opposite, left (within group, Tukey). Bottom panels: representative probe-trial. MWM images modified from ref. [9]; Image credit for brains schematics: Allen Institute.

For retrograde tracing experiment, 0.25 μl of FG (4% in NaCl 0.9%) were injected unilaterally in the VS at −4.3 mm from the dura. The injector was left in place for 5 additional minutes to allow diffusion. Animals were left to recover for at least 7–9 days to allow FG retrograde diffusion.

Adeno-associated viruses (AAV1/2) were used to express the Venus fluorescent protein and human influenza hemagglutinin (HA)-tagged hM4D$_i$ (AAV-Syn::Venus-2A-HA-hM4D$_i$) in the vSUB, a designer Gα$_i$-coupled receptor activated exclusively by CNO[23]. Viral vector was a generous gift of Cornelius T. Gross. The AAV was inoculated bilaterally in the vSUB at −4.3 mm from the dura. The volume of injection was 0.30 μl per side, delivered with the use of glass pipettes. Pipettes were left in place for 5 additional minutes after injection to allow diffusion[39]. Animals were allowed to recover for at least 4 weeks (to allow the expression and anterograde diffusion of the virus). After the recovery period, mice underwent bilateral cannulae implantation in the VS as described above.

**Drugs and in vivo focal injection procedure.** General infusion procedure was performed for all experiments immediately after the end of the last session of training. A total of 0.25 μl of 0.6 μg/μl[6,40] NMDA receptor antagonist D(−)-2-amino-5-phosphonopentanoic acid (AP-5; Sigma-Aldrich, Italy), was bilaterally infused in the brain at the injection rate of 0.125 μl/min. The dose of AP-5 was decided based on previous published data[6,40]. AP-5 was dissolved in phosphate-buffered saline (PBS 0.1 M). Control mice were identically injected with the same volume of PBS. After administration, the injector was left in place for 30 additional seconds to allow diffusion. A total of 0.30 μl of CNO (Cayman Chemical, USA) 10 mM was dissolved in dimethyl sulfoxide (DMSO) and PBS solution (1:1). The dose was chosen based on previous literature[23]. Control mice were injected with the same volume of vehicle solution (DMSO:PBS).

**Behavioral procedures.** The spatial water maze task (sMWM) consisted of three different phases distributed across three consecutive days: Familiarization, Training and Test[41]. Familiarization consisted of one session of three consecutive trials (intertrial interval: 20 s). No cues were attached to the wall and a 9-cm-diameter platform protruded 1.5 cm above the water surface. The session started with the animal standing on the platform for 20 s. Animals were introduced in non-target quadrant in a pseudo-random order facing the wall. If they did not reach the platform in 60 s, they were gently led to the platform by the experimenter. Training consisted of six consecutive sessions (intersession interval: 5 min) of three trials (intertrial interval: 30 s). The procedure was the same as in the familiarization phase except for the platform that was submerged 0.5 cm beneath the surface of the water. The platform position was always different than during familiarization and the pool was surrounded by several extra-maze cues attached to the wall. Test was performed 24 h after the last training session and consisted of a single probe trial. The platform was removed, and mice were allowed a 60-s search for the platform starting from the center of the pool.

In the cue version of the water maze task (cMWM) all the distal cues were removed and a single proximal cue was present, a black-painted ball hanging 22 cm above the hidden platform. The position of the platform and the ball changed across sessions to prevent animals from using spatial bias. During the test, the platform was removed, and the ball was in the quadrant opposite to that used in the last training session.

The ODT apparatus consisted of a circular open field, 60 cm in diameter. A square stripped pattern (30 × 30 cm, alternating 1.5 cm wide vertical white and black bars) was attached to the wall of the open field, always in the same position. Four identical objects were used in this task. The procedure consisted in five sessions[6,42]. Briefly, in session 1 (S1, familiarization), the animals were individually placed in the center of the empty open field for a 6-min session to familiarize with the apparatus. Subjects were then removed and placed back in a holding cage. After 2 min inter-trial interval, they were placed in the open field (with objects) for three consecutive 6-min sessions (training phase sessions 2–4; S2–S4) separated by a 2-min interval, during which the animals were returned to the holding cage. During the training phase, the configuration of the objects was the same and the mice were introduced in the open field facing the wall, always from a different position. Twenty-four hours later, the mice were submitted to the spatial change session (S5) in which two of the objects (displaced objects—DO) were repositioned as compared to previous sessions. Mice were placed in the open field facing the wall, from a novel position never used during training.

**fos and FG co-localization.** For FG-immunofluorescence procedure mice were deeply anesthetized with an overdose dose of anesthetic and transcardially perfused with 40 ml of 0.9% NaCl solution (room temperature), followed by 40 ml of 4% paraformaldehyde solution in PBS (4 °C). Free-floating sections 40 μm thick underwent 1 h blocking step (5% Normal Goat Serum and 0.1% Triton X-100 in PBS) and overnight incubation with primary antibody against fos (1:300; sc-52, Santa-Cruz Biotechnology, USA) at 4 °C, followed by washes and 1 h incubation with fluorescent-labeled secondary antibody (1:400, Rhodamine Red™-X-conjugated goal anti-rabbit, 111-295-144, ImmunoResearch, USA) at room temperature.

A subset of 4 slices was acquired with a fluorescence microscope with a 4× magnification. A mosaic of the ipsi- and contralateral FG-injected hemispheres and

the corresponding vSUB were manually acquired and stitched with FIJI (ImageJ, N.I.H., USA) and analyzed for FG diffusion. Two representative images of the VS and the vSUB were treated for nuclear counterstaining with VECTASHIELD® Antifade Mounting Medium with propidium iodide (PI) (Vector Laboratories, USA). PI-stained slices were acquired at the Olympus iX83-FV1200 confocal laser scanning microscope with a 10× NA 0.40 objective, 559 nm laser/PI setting for PI detection. Single images of 1600 × 1600 pixels were stitched together in a mosaic view with the Multi Area Viewer tool (Olympus Fluoview 4.2).

fos and FG images were acquired with a confocal microscope (Leica TCS-SP5) with a 20×/0.64× objective in the vSUB. FG fluorescence was detected under 405 nm Diode laser, while fos under 543 nm HeNe laser. Approximately four images (1024 × 1024 pixel) with identical settings were acquired from alternated slice per mouse. fos/mm², FG/mm² and co-localization/mm² were analyzed in eight non-overlapping regions of interest (ROIs, 80 × 80 μm) with Imaris 7.6.5 software by an experimenter blind to treatment. Values from images were averaged per subject, and subsequently per group (naïve, cue and spatial).

**Golgi-Cox staining.** Golgi-Cox staining was used to highlight dendritic arborization and spines. To this aim, 24 h after training, naïve, cue and spatial trained mice were deeply anesthetized with an overdose dose of anesthetic and transcardially perfused with 50 ml of 0.9% NaCl at room temperature. Brains were collected and impregnated using a Golgi-Cox solution (1% potassium dichromate, 1% mercuric chloride and 0.8% potassium chromate) for 6 days[43]. vSUB slices were cut for placement verification, which was made on mounted slices without additional staining.

Dendritic spine density analysis was performed on 30–100 μm dendritic segments of MSNs, in both the core and shell of VS. Spine density was measured using the software Neurolucida (Microbrightfield, USA) connected to an optical microscope (Leica, DMLB). Neurons were first identified under low magnification (20×/NA 0.5); subsequently, spine analysis was performed only in neurons that were (i) relatively isolated from neighboring neurons, (ii) consistently and darkly impregnated along the entire extent of the dendrites and (iii) had untruncated dendrites. Three to nine neurons per animal (depending on the quality of the staining) were analyzed. Spines were quantified under high magnification (100×/ NA 1.5) along the entire branching, using a camera connected to the microscope (Qimaging Qicam Fast1394, Canada). Only protuberances with clear connection of the head of the spine to the shaft of the dendrite were counted as spines. The number of spines per neuron from each individual were averaged and counted as single sample, and subsequently the final values per animal were averaged per group (naïve, cue and spatial). Values for cue and spatial trained mice were normalized on naïve values separately for each replicate of the experiment. All measurements were performed by an experimenter blind to the experimental conditions.

**Acute slice recordings.** Five to eight weeks after AAV inoculations in vSUB, mice were deeply anesthetized with an intraperitoneal injection of an overdose of anesthetic (a mixture of tiletamide/zolazepam, zoletil 80 mg/kg, and xilazine 10 mg/kg) and transcardially perfused with ice-cold, artificial cerebrospinal fluid (ACSF). Brains were sectioned at the level of the VS in 250-μm-thick coronal sections on a VT-1200 vibratome (Leica, Germany). The ACSF used to both perfuse the animals and slice the brains was modified to contain (in mM): 75 sucrose, 87 NaCl, 2.5 KCL, 1.25 NaH₂PO₄, 7 MgCl₂, 0.5 CaCl₂, 26 NaHCO₃, 11 glucose and 3 sodium ascorbate. Slices were placed in a holding chamber filled with room temperature ASCF saturated with 95% O₂ and 5% CO₂ containing (in mM): 125 NaCl, 2.5 KCl, 1.25 NaH₂PO₄, 1 MgCl₂, 2.4 CaCl₂, 26 NaHCO₃, 11 glucose and 1 sodium ascorbate (310–315 mOsm). After incubation for at least 45 min, an individual slice was transferred to a submerged recording chamber and continuously superfused at 33–34 °C with oxygenated ACSF at a rate of 3–4 ml/min. The ACSF used in the recording chamber was the same except for the exclusion of sodium ascorbate and, to block inhibitory transmission, the addition of the GABA$_A$ receptor channel blocker, picrotoxin (100 μM, Tocris, UK). Picrotoxin was dissolved in DMSO. The final concentration of DMSO in the bathing solution was 0.1%. At this concentration, DMSO alone did not modify the membrane potential, input resistance or firing properties of VS neurons.

Borosilicate pipettes (WPI, USA, 1.5 mm OD × 0.86 mm ID) were pulled with a micropipette puller (P97Sutter Instruments, USA), to obtain a resistance ranging from 4 to 5 MΩ. Pipettes were filled with an internal solution containing (in mM): 70 K-gluconate, 70 KCl, 10 HEPES, 4 EGTA, 2 NaCl, 4 Mg-ATP and 0.3 Na-GTP, 5 Na-phosphocreatine (295–300 mOsm). Cells were visualized with 60× water-immersed objective mounted on upright microscope (Nikon, eclipse FN1) equipped with a CCD camera (Scientifica, UK). Whole-cell voltage-clamp recordings were performed using a MultiClamp 700B amplifier (2 kHz low-pass Bessel filter and 10 kHz digitization) with pClamp 10.3 software (Molecular Devices, USA). Venus⁺ terminals expressing hM4D$_i$ were visualized by a 473 nm LED (pE1, Cool LED, UK). MSNs in the medial VS were identified by morphology, and hyperpolarized resting membrane potential. Series resistance (10–25 MΩ) was monitored with a 5 mV hyperpolarizing pulse (50 ms) given every 30 s, and cells exhibiting 20% changes were discarded. MSNs were voltage clamped at −80 mV and spontaneous glutamatergic post synaptic currents (sEPSCs) were recorded in control conditions and during perfusion with CNO (10 μM). Experiments in

current clamp configuration were performed on the same spiny neurons in order to evaluate their resting membrane potentials ($V_m$), input resistance ($R_{input}$) and firing mode in control conditions and during perfusion with CNO (10 μM). Repetitive hyperpolarizing and depolarizing current pulses (800 ms duration, 50 pA incremental amplitude) were delivered every 2 s through patch pipette at resting membrane potential. Measurements of membrane potentials were not corrected for the liquid junction potential error. Drugs were applied in the bath by gravity by changing the perfusion solution to one differing only in its content of drug(s). The ratio of flow rate to bath volume ensured complete exchange within 2–3 min.

EPSCs were analyzed using pClamp software, which uses a detection algorithm based on a sliding template. The template did not induce any bias in the sampling of events because it was moved along the data trace by one point at a time and was optimally scaled to fit the data at each position. The detection criterion was calculated from the template-scaling factor and from how closely the scaled template fitted the data. Detection parameters were set at an amplitude >4 pA, and acquired events were visually inspected before averaging. A minimum of 100 sEPSCs was analyzed during 3 min of the baseline period and 3 min during CNO perfusion. $R_{input}$ values were obtained from $\Delta V$ measured in response to hyperpolarizing current steps (800 ms duration, −50 pA).

Additional methods are given in the Supplementary information.

**Reporting summary**. Further information on research design is available in the Nature Research Reporting Summary linked to this article.

## Data availability
The data that support the findings of this study are available from the corresponding author upon reasonable request.

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

## Acknowledgements
The authors would like thank Cornelius T. Gross and Enrico Cherubini for critically reading the manuscript and for the stimulating discussions, Arianna Leonardo for her help with the experiments, and Cornelius Gross and Angelo Raggioli for providing the AAVs and expertise in chemogenetics. This study was supported by grant from the University of Rome "La Sapienza" (to A.M.; A.R.), and from a NARSAD independent investigator grant (to A.M.).

## Author contributions
G.T. carried out, designed and analyzed all the experiments, except for the electrophysiology experiments, which were carried out and analyzed by M.G.; chemogenetics experiments were tested by L.A., V.K., V.M. and E.C.; A.P. collaborated for the spine analyses; G.M.B. and L.A. for the pharmacology experiments, V.d.T. for the virus expression analysis. The project was conceived by G.T. and A.M. with critical input from A.R. and M.A.-T. The manuscript was written by G.T. and A.M. with inputs from A.R. and M.A.-T.

## Competing interests
The authors declare no competing interests.
