## [Peer Review File · Nature Communications]

Reviewers' Comments:

Reviewer #1:

Remarks to the Author:

The authors studied how communication between the hippocampus and ventral striatum contribute to recall of spatial memories. First, the authors identified activated projections from the hippocampus, through the subiculum and into the ventral striatum for spatial memory recall. They also impaired communication between the subiculum and striatum by means of pharmacological disconnection. This was done immediately after learning and their results suggest the importance of post learning communication between the two brain areas for later memory recall. They also provide evidence of structural changes in the striatum that occur after initial learning. It is also shown that selective inhibition of striatum projecting neurons in the subiculum decreases EPSC. Overall, this work provides strong evidence for the importance of the hippocampal striatal network for successful learning and later memory recall. This is a strong series of experiments that is very well focused and well controlled. I believe this will be a significant addition to the literature, but there are some presentation problems that detract from its potential impact.

All of my concerns are minor in nature, but on the whole there are many problems with the language used. Many instances with inconsistent tense, poor application of plural forms of words, a lack of commas, and overall poor flow to the writing. Again, all of these are minor concerns, but they add together in a way that certainly takes the reader's concentration away from the science.

Here are some concerns listed in the order they appear in the text. (this list is not comprehensive, please, try to revise the entire manuscript with these types of language changes in mind).

L40 – the phrasing of “might be rather sustained” seems like something is missing.

L51 – this sentence reads strangely – change.

L55 – the verb “sustain” seems wrong here.

L 97 – start new paragraph

L103 – “this” should be “these”

L140 – the verb “posing” seems like a poor choice here.

L143 – “experiment” – add an “s” and a comma after “particular”

L146 – “experiment” – add an “s”

L147 – “manipulation” – add an “s”

L147 – “suggests” instead of “suggesting”

L148 – “when actual experience is ended” is awkward.

L151-2 – the circuitry is not clearly described here – rework these sentences.

L153 – add comma after “Interestingly” and add “s” to “projection”

L154 – replaces “exerts” with “leads to”

L165 – this sentence is awkward

Define acronyms in figure 2 – also, please use only one symbol (preferably asterisks) for indicating significance at a certain p value. It's currently distracting and confusing with so many symbols.

Reviewer #2:

Remarks to the Author:

Torromino et al. report that cross regional communication between the ventral subiculum (vSUB) and ventral striatum (VS) is necessary for spatial memory consolidation in a Morris Water Maze (MWM). Mice with selective pharmacological inactivation of the hippocampus (HPC)-VS pathway showed

learning deficits selectively in a spatial MWM, but not cue-based MWM. They report increased VS spine density 24 hr after spatial MWM training but not cued MWM training. Supporting their claim that vSUB-VS interactions are required for spatial MWM consolidation, the increase in VS spine density after training was dependent on vSUB-activity. Finally, they report that reducing the activity of VS-targeting glutamatergic vSUB neurons (using DREADDs) leads to spatial memory impairment in the spatial MWM. This provides direct evidence that projections between the two brain regions, specifically glutamatergic vSUB neurons projecting to VS, are necessary for spatial memory consolidation. However, as mentioned in the introduction, disconnecting the HPC-nucleus accumbens shell pathway has previously been shown to impair context-based spatial memory (Ito et al. 2008 J Neurosci). Thus the novelty of the work must be clarified in light of this prior work. Furthermore, while the results as presented are promising, there are concerns dampening the claims, described below.

Major concerns:

Since there is growing evidence that glutamatergic input from vSUB modulates dopamine (DA) function in VS (Grace AA 2016 Nat Rev Neurosci), it is important to note that the implications of vSUB-VS functional disconnection could be partially due to changes in DA release, or effects of DA on novelty and learning.

- Dopamine receptor or reuptake blockers should be used to determine if the observed reduction of medial spiny neuron activity in the VS and impairment in sMWM following CNO are due to changes in DA.
- Control experiments are necessary to compare animals' baseline and post-administration motivation levels and novelty preference. Sucrose preference or conditioned place preference should be used to test reward-related behavior and cue-place associative learning.
- Furthermore, current literature suggests that the vSUB-nucleus accumbens shell projection contributes to cocaine-induced reinstatement of drug seeking behavior. There should be a discussion of HPC-VS connections implicated in learning and long-term memory broadly, not just spatial memory.

Behavior controls for overall activity levels, motivation (commented on above), and anxiety-like behavior are lacking.

- Open field or elevated plus maze should be included to measure anxiety-like behavior.
- A random foraging task or similar task should be used to measure overall locomotion (speed, distance traveled, etc).

The nonspecific effects of this DREADD manipulation and the time course over which it lasts are unclear. As a result it is unclear when vSUB-VS interactions are essential for MWM performance.

- How well characterized is this particular DREADD in these brain regions?
- How long do CNO's effects remain and do they change over time? This is especially important because silencing can lead to homeostatic effects. In slices, experiments should determine how long CNO remains effective if present in the bath and how long it takes for CNO's effects to wash out after it is no longer present in the bath.
- There is no mention of this DREADDs' effects on activity in vivo and how long after the infusion the effects last. Therefore it is unclear when vSUB-VS interactions are essential for this task. Are they key right after training? For hours afterward? MWM experiments should be done with different latencies between training and CNO administration to determine the key period for vSUB-VS interactions.

Acute slice recording results are showing only 7 out of 13 cells recorded. There is no indication of how many animals these cells were from.

- The results should be based on more cells, like 10-15 cells from 3-5 different mice. More would be better.
- Data from all recorded cells on changes in frequency, sEPSC amplitudes, membrane potentials, input resistance, bursting probability/interburst interval, latency to response, as well as, the number of

animals used should be reported.

- Figure 4f: Are the labels supposed to be vehicle vs. CNO?

Minor concerns:

The mention of encoding in the introduction section is misleading since all of the behavioral data are shown in animals extensively trained prior to pharmacologic manipulations and encoding of spatial information is not directly tested in this work.

Additionally, the rationale for targeting the vSUB-VS pathway, and not others, should be clearly explained.

Histological images (like Figure 1b,c) should include DAPI or similar stain to see the overall brain architecture.

Figure legend for Figure 1 should explain what the red and yellow arrows are.

There are a host of grammatical errors, typos, and other issues in this manuscript with examples listed below:

Main paper

- Figure 4d caption does not match the figure - shows cumulative probability of ISI, not the sEPSCs amplitude.
- I would like to see the individual data points and their distribution for each bar graph.
- There is no quantification of the extent of viral infusion.
- Figure 1b - DAPI should be used to visualize the architecture.

Supplementary

- Figure S4 is the same as S3. I'm assuming the S4 caption is supposed to be for injection sites in the functional disconnection and spatial MWM experiments?
- Typo on S5 - d. "Reaning"
- No number of animals indicated in S6. It would be helpful to have a table showing the number of mice used per experiment along with the corresponding figure number, pharmacologic reagent, and type of behavioral task (spatial vs cue).
- I had a hard time understanding the different color schemes in figure S10. Are different colors other than green and red necessary? Either stick to one color for overlapped regions, or clearly label what each of the colors means.
- Figure S11 - g. Typo again "Reaning"
- Figure S12 - might want to change the black squares to some other color because they are hard to see.

Reviewer #3:

Remarks to the Author:

Nature Communications review 2018

I have reviewed the paper carefully and it has some really interesting elements and the issue of interactions between the hippocampus and ventral striatum is certainly an important one. They have provided some evidence for a role of this circuit is involved in memory consolidation of spatial memory but it is not completely convincing because of some controls that were missing. I would also like to see an integration of the work with recent theoretical ideas that the ventral hippocampal/VS circuit is important for early triaging of spatial behaviours which would fit nicely with their behavioural paradigm and results.

My specific comments can be found below.

INTRODUCTION:

I found the conceptualizations presented in the introduction a bit difficult to understand. For example, I am not sure about what they mean this wide-spread network they describe is involved in encoding? Does this mean it is different than retrieval?

A relevant view for the present paper is one that suggests that that the glutamate projection to the VS is probably an output for early triaging of behaviour (Gruber and McDonald, 2012) to get the animal over to the general area the escape platform is located to generate more precise instrumental behaviours. This view nicely fits with some of their data (disconnections) and their accelerated and rapid training procedure.

The role of the ventral striatum in memory consolidation is based on post-training manipulations and some of the current experiments use this procedure but they do not do an important control. The traditional experiments have a control in which the same manipulation is done but several hours after the training experience. In this instance, there should be no impairment on the spatial task with NMDA receptors blocked in the purported neural circuit.

METHODS:

I liked the use of the ipsilateral and contralateral manipulations.

-for the virus injection in ventral subiculum this is AAV but this is really a DREADD experiment. Why not say so? My biggest concern with this manipulation and the interpretation of the results is that no CNO on its own is reported in the design or results. This is critical as there have been recent reports of CNO and clozapine having functional effects and in some cases this might account for the entire effect.

-spelling on line 223 (open field)

-for the injections of AP5 (ipsi and contralateral) the researchers used a variant of the water maze-three phases of training and there may be some strengths and weaknesses to their approach. During the familiarization phase, even if no cues are present, there is little doubt that the hippocampal formation forms a spatial representation very quickly including head direction and probably grid cells. The training is all done in one day, which is fine depending on your research question, but I would like to see some traditional measures of the acquisition presented including graphs of latency, heading direction or some kind of corridor analysis, etc. I see path length in the supplemental material but would like to see how good they are. This will give the reader an idea how good the animals are at finding the platform. Based on our experience, this amount of training the mice will not be very good and not showing much spatial specificity.

On the test, 24 hours after the training I would also like to see quadrant preference data in terms of latency and some spatial specificity measures (first quadrant entered, annulus crossings) including swim paths.

Also, in terms of the cued trials, there is little doubt that the hippocampal formation acquires spatial information during cue training but it is not necessary for accurate performance (see early Morris behaviour experiments using cue and then spatial probes as well as Sutherland and Rudy, 1988; McDonald and White, 1994) which may help explain some of your data.

CFOS AND FLUOROGOLD STAINING:

For these manipulations why does this not include the non-biased stereology methods.

GOLGI STAINING AND DENDRITES

-in cued versus spatial trained rats. What is the prediction here? I am assuming hippocampus forms a representation during cue training but maybe the output circuits actually controlling behaviour are different (other subcircuits in the medial striatum for example-see Gruber and McDonald, 2012) and that is why you don't see dendritic changes in VS in these training conditions. If this is the logic what does this mean for the type of information being consolidated in VS in these conditions?

ASSEMENT OF MEDIUM SPINY NEURONS IN THE VS CORE AND SHELL

The number of spines per neuron is one measure but others would have been of interest (see paper by Kolb and colleagues).

We would like to thank the editor and the reviewers for their careful work and the useful suggestions on the previous versions of the manuscript.

Please find below a point-by-point answer (black) to the reviewers' concerns.

REVIEWER # 1:

The authors studied how communication between the hippocampus and ventral striatum contribute to recall of spatial memories. First, the authors identified activated projections from the hippocampus, through the subiculum and into the ventral striatum for spatial memory recall. They also impaired communication between the subiculum and striatum by means of pharmacological disconnection. This was done immediately after learning and their results suggest the importance of post learning communication between the two brain areas for later memory recall. They also provide evidence of structural changes in the striatum that occur after initial learning. It is also shown that selective inhibition of striatum projecting neurons in the subiculum decreases EPSC. Overall, this work provides strong evidence for the importance of the hippocampal striatal network for successful learning and later memory recall. This is a strong series of experiments that is very well focused and well controlled. I believe this will be a significant addition to the literature, but there are some presentation problems that detract from it's potential impact.

All of my concerns are minor in nature, but on the whole there are many problems with the language used. Many instances with inconsistent tense, poor application of plural forms of words, a lack of commas, and overall poor flow to the writing. Again, all of these are minor concerns, but they add together in a way that certainly takes the reader's concentration away from the science.

Here are some concerns listed in the order they appear in the text. (this list is not comprehensive, please, try to revise the entire manuscript with these types of language changes in mind).

L40 – the phrasing of “might be rather sustained” seems like something is missing.

L51 – this sentence reads strangely – change.

L55 – the verb “sustain” seems wrong here.

L 97 – start new paragraph

L103 – “this” should be “these”

L140 – the verb “posing” seems like a poor choice here.

L143 – “experiment” – add an “s” and a comma after “particular”

L146 – “experiment” – add an “s”

L147 – “manipulation” – add an “s”

L147 – “suggests” instead of “suggesting”

L148 – “when actual experience is ended” is awkward.

L151-2 – the circuitry is not clearly described here – rework these sentences.

L153 – add comma after “Interestingly” and add “s” to “projection”

L154 – replaces “exerts” with “leads to”

L165 – this sentence is awkward

We would like to thank this referee for the careful reading of our manuscript. Because the introduction has been entirely re-written, we used the rephrasing suggestions to avoid errors in the revised version.

Define acronyms in figure 2 – also, please use only one symbol (preferably asterisks) for indicating significance at a certain p value. It's currently distracting and confusing with so many symbols.

Acronyms in Fig.2 have been defined, we also reduced as much as possible the symbols for indicating significance: * within groups, # between groups.

REVIEWER # 2:

1. Torromino et al. report that cross regional communication between the ventral subiculum (vSUB) and ventral striatum (VS) is necessary for spatial memory consolidation in a Morris Water Maze (MWM). Mice

with selective pharmacological inactivation of the hippocampus (HPC)-VS pathway showed learning deficits selectively in a spatial MWM, but not cue-based MWM. They report increased VS spine density 24 hr after spatial MWM training but not cued MWM training. Supporting their claim that vSUB-VS interactions are required for spatial MWM consolidation, the increase in VS spine density after training was dependent on vSUB-activity. Finally, they report that reducing the activity of VS-targeting glutamatergic vSUB neurons (using DREADDs) leads to spatial memory impairment in the spatial MWM. This provides direct evidence that projections between the two brain regions, specifically glutamatergic vSUB neurons projecting to VS, are necessary for spatial memory consolidation. However, as mentioned in the introduction, disconnecting the HPC-nucleus accumbens shell pathway has previously been shown to impair context-based spatial memory (Ito et al. 2008 J Neurosci). Thus the novelty of the work must be clarified in light of this prior work. Furthermore, while the results as presented are promising, there are concerns dampening the claims, described below.

We thank the referee for pointing this out this aspect. We think that the revised version, especially paragraphs 3 and 4 of the introduction, definitely emphasizes the novelty/originality of our work.

2.Since there is growing evidence that glutamatergic input from vSUB modulates dopamine (DA) function in VS (Grace AA 2016 Nat Rev Neurosci), it is important to note that the implications of vSUB-VS functional disconnection could be partially due to changes in DA release, or effects of DA on novelty and learning.

- Dopamine receptor or reuptake blockers should be used to determine if the observed reduction of medial spiny neuron activity in the VS and impairment in sMWM following CNO are due to changes in DA.
- Control experiments are necessary to compare animals' baseline and post-administration motivation levels and novelty preference. Sucrose preference or conditioned place preference should be used to test reward-related behavior and cue-place associative learning.
- Furthermore, current literature suggests that the vSUB-nucleus accumbens shell projection contributes to cocaine-induced reinstatement of drug seeking behavior. There should be a discussion of HPC-VS connections implicated in learning and long-term memory broadly, not just spatial memory.

The referee raised a crucial point regarding the possible role of dopamine (DA) in mediating the effects of vSUB-VS projection on spatial memory consolidation. The role of striatal DA and of DA x glutamate interactions in spatial memory formation has been extensively demonstrated by us (Coccarello et al., *Psychopharmacology*, 152(2), 189–99, 2000; Mele et al., *Behavioural pharmacology*, 15(5-6), 423–31, 2004; Ferretti et al., *Psychopharmacology*, 179, 108–116, 2005; Coccarello et al., *Neuropsychopharmacology* : 37(5), 1122–1133, 2012) and others (Ploeger et al., *Behav Neurosci* 108:927–934, 1994; Setlow and McGaugh, *Behav Neurosci* 112:603–610, 1998). Although it could be of interest to know how and to which extent DA transmission is involved in the effect we observed, the overall goal of our study was to define the functional role of specific anatomical connections between the vSub and VS in mediating spatial memory consolidation.

3.Behavior controls for overall activity levels, motivation (commented on above), and anxiety-like behavior are lacking.

- Open field or elevated plus maze should be included to measure anxiety-like behavior.
- A random foraging task or similar task should be used to measure overall locomotion (speed, distance traveled, etc).

We run new experiments shown in Fig.S12 (pg. 7; ln. 3-4) which confirm that DREADDs-mediated inhibition of the vSUB-VS pathway does not interfere with motor activity, motivation and anxiety.

4.The nonspecific effects of this DREADD manipulation and the time course over which it lasts are unclear. As a result it unclear when vSUB-VS interactions are essential for MWM performance.

- How well characterized is this particular DREADD in these brain regions?
- How long do CNO's effects remain and do they change over time? This is especially important because silencing can lead to homeostatic effects. In slices, experiments should determine how long CNO remains effective if present in the bath and how long it takes for CNO's effects to wash out after it is no longer present in the bath.

- There is no mention of this DREADDs' effects on activity in vivo and how long after the infusion the effects last. Therefore it is unclear when vSUB-VS interactions are essential for this task. Are they key right after training? For hours afterward? MWM experiments should be done with different latencies between training and CNO administration to determine the key period for vSUB-VS interactions.

We agree with this referee (see also referee 3, point 1) that the time-window of the disconnection effect needs to be clearly established since we hypothesize that the vSub-VS pathway activated shortly post-training controls the early phase of spatial memory consolidation. Accordingly, we run a new experiment to silence this pathway 120' post-training (Fig.4h; Fig.S11d,f; from pg.6; ln26 to pg.7 ln3). The observation that late silencing did not produce any impairing effect on memory reinforces our hypothesis that the vSub-VS pathway is critical only for the early phase of spatial memory consolidation.

We should also point out that in the electrophysiological experiments, coherently with previous reports (Stachniak et al., 2014; Roth, 2016), we obtained a partial recovery of EPSCs frequency (61% of EPSCs frequency before CNO application) in 5 out of 13 cells 6.4 ± 0.6 min after CNO (data not shown).

5. Acute slice recording results are showing only 7 out 13 cells recorded. There is no indication of how many animals these cells were from.

In the revised version, the number of animals used for slice recording has been included in the results section (pag. 6, line 12 and 20; legend Figure 4 and S9) (see also next point).

5.1 The results should be based on more cells, like 10-15 cells from 3-5 different mice. More would be better.

Patch clamp experiments in slices have been repeated using a potassium-based intracellular solution to better characterize the neuronal cell type from its electrical properties. Results obtained confirm the CNO induced reduction in firing frequency observed in the original experiments. Excitatory postsynaptic currents and intrinsic properties have been recorded in larger samples (13 spiny neurons from 6 mice) (pag. 6, line 12 and 20; legend Figure 4 and S9 and Table S1).

5.2 Data from all recorded cells on changes in frequency, sEPSC amplitudes, membrane potentials, input resistance, bursting probability/interburst interval, latency to response, as well as, the number of animals used should be reported.

Data on changes in frequency and amplitude have been included in Figure 4b-d; data on membrane potentials, input resistance, firing frequency have been included in supplementary Figure 9a-d; latency to response has been added in the main text (result section, pag. 6, line 12 and 20).

Minor concerns:

1.The mention of encoding in the introduction section is misleading since all of the behavioral data are shown in animals extensively trained prior to pharmacologic manipulations and encoding of spatial information is not directly tested in this work.

The introduction has been entirely revised.

2.Additionally, the rationale for targeting the vSUB-VS pathway, and not others, should be clearly explained. This has now been explained (pag. 3; line 11-18).

3.Histological images (like Figure 1b,c) should include DAPI or similar stain to see the overall brain architecture.

Histological images have been replaced with new images counterstained with the nuclear dye propidium iodide (Fig1b).

4.Figure legend for Figure 1 should explain what the red and yellow arrows are.

Legend for Figure 1 has been corrected.

There are a host of grammatical errors, typos, and other issues in this manuscript with examples listed below:

Main paper:

- Figure 4d caption does not match the figure - shows cumulative probability of ISI, not the sEPSCs amplitude.

Figure 4d has been emended, we apologize for the error.

- I would like to see the individual data points and their distribution for each bar graph.

Individual data points have been included in all figures.

- There is no quantification of the extent of viral infusion.

The extent of virus expression has now been included in Fig.S10 (see legend).

- Figure 1b - DAPI should be used to visualize the architecture.

Slices have been counterstained with a nuclear dye, images have been changed.

Supplementary

- Figure S4 is the same as S3. I'm assuming the S4 caption is supposed to be for injection sites in the functional disconnection and spatial MWM experiments?

We apologize, the mistake has been corrected.

- Typo on S5 - d. "Reaning"

Corrected, now Fig.S5.

- No number of animals indicated in S6. It would be helpful to have a table showing the number of mice used per experiment along with the corresponding figure number, pharmacologic reagent, and type of behavioral task (spatial vs cue).

Supplementary table1 now summarizes for each experiments the number of mice, the treatment, the reagent, the task, and the number of the figure where the results of each experiment are shown.

- I had a hard time understanding the different color schemes in figure S10. Are different colors other than green and red necessary? Either stick to one color for overlapped regions, or clearly label what each of the colors means.

We changed the figure and revised the legend, we hope now it is more clear to readers.

- Figure S11 - g. Typo again "Reaning"

Corrected now Fig.S13.

- Figure S12 - might want to change the black squares to some other color because they are hard to see.

We revised all figures with the injection sites. We hope it is now clearer to readers.

REVIEWER # 3

I have reviewed the paper carefully and it has some really interesting elements and the issue of interactions between the hippocampus and ventral striatum is certainly an important one. They have provided some evidence for a role of this circuit is involved in memory consolidation of spatial memory but it is not completely convincing because of some controls that were missing. I would also like to see an integration of the work with recent theoretical ideas that the ventral hippocampal/VS circuit is important for early triaging of spatial behaviours which would fit nicely with their behavioural paradigm and results.

My specific comments can be found below.

INTRODUCTION:

I found the conceptualizations presented in the introduction a bit difficult to understand. For example, I am not sure about what they mean this wide-spread network they describe is involved in encoding? Does this mean it is different than retrieval?

A relevant view for the present paper is one that suggests that that the glutamate projection to the VS is probably an output for early triaging of behaviour (Gruber and McDonald, 2012) to get the animal over to the general area the escape platform is located to generate more precise instrumental behaviours. This view nicely fits with some of their data (disconnections) and their accelerated and rapid training procedure.

The introduction has been widely revised (see also referee 2 point 1).

The role of the ventral striatum in memory consolidation is based on post-training manipulations and some of the current experiments use this procedure but they do not do an important control. The traditional experiments have a control in which the same manipulation is done but several hours after the training experience. In this instance, there should be no impairment on the spatial task with NMDA receptors blocked in the purported neural circuit.

We completely agree with the referee the time course of the effect of vSUB-VS silencing on memory consolidation is important (see also referee 2 point 4), a new experiment has been added to answer this question (Fig.4h; Fig.S11d,f; from pg.6; ln26 to pg.7 ln3).

METHODS:

I liked the use of the ipsilateral and contralateral manipulations.

-for the virus injection in ventral subiculum this is AAV but this is really a DREADD experiment. Why not say so? My biggest concern with this manipulation and the interpretation of the results is that no CNO on its own is reported in the design or results. This is critical as there have been recent reports of CNO and clozapine having functional effects and in some cases this might account for the entire effect.

This is also an important point as an effect of CNO by itself can be a concern. Two experiments which rule out this possibility were already included in the original version of the manuscript (FigsS13a-c; S13d-i). In the first experiment, it was shown that focal administration of CNO in the VS does not affect the training performance in the MWM (FigsS13a-c). In the second experiment, it was shown that CNO does not affect locomotor activity and anxiety (FigsS13d-i) (text: pg. pg. 7; lines 5-6).

-Spelling on line 223 (open field)

Text has been rephrased.

-for the injections of AP5 (ipsi and contralateral) the researchers used a variant of the water maze-three phases of training and there may be some strengths and weaknesses to their approach.

During the familiarization phase, even if no cues are present, there is little doubt that the hippocampal formation forms a spatial representation very quickly including head direction and probably grid cells.

The training is all done in one day, which is fine depending on your research question, but I would like to see some traditional measures of the acquisition presented including graphs of latency, heading direction or some kind of corridor analysis, etc. I see path length in the supplemental material but would like to see how good they are. This will give the reader an idea how good the animals are at finding the platform. Based on our experience, this amount of training the mice will not be very good and not showing much spatial specificity.

On the test, 24 hours after the training I would also like to see quadrant preference data in terms of latency and some spatial specificity measures (first quadrant entered, annulus crossings) including swim paths.

Also, in terms of the cued trials, there is little doubt that the hippocampal formation acquires spatial information during cue training but it is not necessary for accurate performance (see early Morris behaviour experiments using cue and then spatial probes as well as Sutherland and Rudy, 1988; McDonald and White, 1994) which may help explain some of your data.

We thank the referee for pointing out the need for more variables to strengthen the validity of our findings. This point was not addressed in the first version of the manuscript to avoid redundancy in the results section

but we agree that adding more variable will help to reinforce our conclusions. Accordingly, the following variables have been added (i) first bearing (head direction) and annulus crossings of mice during probe test; (ii) latency to reach the platform during training. We also show group representative swimming tracks These data are shown in: Fig.2a;2b;2d;4g;4h and in supplementary Figs.S1e;S3b; S3c; S3e; S3f; S6b,c;S11b;c.

CFOS AND FLUOROGOLD STAINING:

6.For these manipulations why does this not include the non-biased stereology methods.

In our study we commonly used standard automatic cell counting (IMARIS software) blind to the treatment (pg. 13; In 22-23).

GOLGI STAINING AND DENDRITES

-in cued versus spatial trained rats. What is the prediction here? I am assuming hippocampus forms a representation during cue training but maybe the output circuits actually controlling behaviour are different (other subcircuits in the medial striatum for example-see Gruber and McDonald, 2012) and that is why you don't see dendritic changes in VS in these training conditions. If this is the logic what does this mean for the type of information being consolidated in VS in these conditions?

The cMWM experiment was performed to control the specific role of vSub-vS projections on spatial memory consolidation. Considering that pharmacological disruption (AP5) of this connection does not interfere with the cMWM performance and that no dendritic spines were formed in the VS, we are confident that the role of these connections is highly specific for spatial memory consolidation.

ASSEMENT OF MEDIUM SPINY NEURONS IN THE VS CORE AND SHELL

The number of spines per neuron is one measure but others would have been of interest (see paper by Kolb and colleagues).

There is good evidence that learning-induced spines basically depicts the degree of structural plasticity in specific regions/circuits. Regarding possible core - shell differences in spatial learning induced changes in spine density could have been very relevant because they could provide a better analysis of the circuits involved. Ongoing experiments in the lab are now trying to address this issue in more detail.

Reviewers' Comments:

Reviewer #1:

Remarks to the Author:

For the most part the authors have addressed all of my previous concerns, however, with the inclusion of all these Supplementary figures for the different controls more needs to be done to explain the need for each control. A sentence or two describing the rationale for the control and how the result addressed that rationale.

In the last sentence of the Discussion and an "s" to "part".

Reviewer #2:

Remarks to the Author:

Torromino et al. have performed significant revisions and the manuscript is considerably improved. Most of my concerns have been addressed. I have one remaining major concern:

The authors added several behavioral assessments, specifically an elevated plus maze, an open field, and a novel object assay. They find no effects of CNO in VS on the aspects of these behavioral assays they measured. They then claim "VS CNO administrations in independent groups of mice demonstrated that activation of hM4Di in VS-projecting subicular terminals was void of effects on anxiety, locomotion and novelty (Fig.S12)" in the Results and state the lack of a significant difference in the distance traveled in the open field "indicates no effects on exploratory behavior and locomotion" in the supplement. First off, this is a strong claim from non-significant results and a few measures. There may be subtle effects not captured by these behavioral measures. The authors should be more careful in interpreting these. Non-significant results do not indicate no effects. Instead they should state they found no significant effects. Furthermore, these additional behaviors do not control for differences in motivation and the authors do not discuss this in the manuscript. This is an important behavioral control as one could argue differences in MWM performance could be due motivation changes. It is possible that the cued MWM tasks controls for differences in motivation. If the authors believe this to be true, it should be discussed.

I also have one minor comment: Showing the individual data points for all of the bar graphs is very helpful. Annulus frequency in the target has a wide distribution of data, wider than other groups (eg.. Supp Fig. 3c,f, Supp. Fig. 6, Supp. Fig. 11c,d). Please discuss why this is. Does this mean some animals did not learn or something else?

Reviewer #3:

Remarks to the Author:

I am satisfied with the changes made to the manuscript based on my comments.

Please find below a point-by-point answer (black) to the reviewers' concerns.

REVIEWER # 1:

Reviewer #1 (Remarks to the Author):

For the most part the authors have addressed all of my previous concerns, however, with the inclusion of all these Supplementary figures for the different controls more needs to be done to explain the need for each control. A sentence or two describing the rationale for the control and how the result addressed that rationale.

The rationale and the results for the 120 min post-training experiment can be found respectively on page 7 line 6-8 and on page 7 line 10-11. The rationale and the results for the EPM/OF/Novelty experiment can be found at page 7 line 10-14.

In the last sentence of the Discussion and an "s" to "part".

This has been changed (pg. 9, ln. 13)

REVIEWER # 2:

Torromino et al. have performed significant revisions and the manuscript is considerably improved. Most of my concerns have been addressed. I have one remaining major concern:

The authors added several behavioral assessments, specifically an elevated plus maze, an open field, and a novel object assay. They find no effects of CNO in VS on the aspects of these behavioral assays they measured. They then claim "VS CNO administrations in independent groups of mice demonstrated that activation of hM4Di in VS-projecting subicular terminals was void of effects on anxiety, locomotion and novelty (Fig.S12)" in the Results and state the lack of a significant difference in the distance traveled in the open field "indicates no effects on exploratory behavior and locomotion" in the supplement. First off, this is a strong claim from non-significant results and a few measures. There may be subtle effects not captured by these behavioral measures. The authors should be more careful in interpreting these. Non-significant results do not indicate no effects. Instead they should state they found no significant effects.

The text (page 7, line 12) and the legend to supplementary figure 12 (page 23, line 8 and 10) have been changed accordingly to the reviewer suggestion.

Furthermore, these additional behaviors do not control for differences in motivation and the authors do not discuss this in the manuscript. This is an important behavioral control as one could argue differences in MWM performance could be due motivation changes. It is possible that the cued MWM tasks controls for differences in motivation. If the authors believe this to be true, it should be discussed.

This discussion has now been included (Page 5, line 18-20).

I also have one minor comment: Showing the individual data points for all of the bar graphs is very helpful. Annulus frequency in the target has a wide distribution of data, wider than other groups (eg.. Supp Fig. 3c,f, Supp. Fig. 6, Supp. Fig. 11c,d). Please discuss why this is. Does this mean some animals did not learn or something else?

In our experience with the MWM, annulus frequency has always been more variable. Nevertheless the result has the same trend of the other variables (distance travelled and time spent in the quadrants).

REVIEWER # 3

I am satisfied with the changes made to the manuscript based on my comments.